

# A global dataset of daily near-surface air temperature at 1-km resolution (2003-2020)

Tao Zhang[1], Yuyu Zhou[1], Kaiguang Zhao[2], Zhengyuan Zhu[3], Gang Chen[4], Jia Hu[1], Li Wang[5]

5   [1] Department of Geological and Atmospheric Sciences, Iowa State University, Ames, IA, 50011, USA

[2] School of Environment and Natural Resources, Ohio Agricultural Research and Development Center, The Ohio State University, Wooster, OH, 44691, USA

[3] Department of Statistics, Iowa State University, Ames, IA, 50011, USA

[4] Laboratory for Remote Sensing and Environmental Change (LRSEC), Department of Geography and Earth Sciences,
10   University of North Carolina at Charlotte, Charlotte, NC, 28223, USA

Department of Statistics, George Mason University, Fairfax, VA, 22030, USA

*Correspondence to:* Yuyu Zhou (yuyuzhou@iastate.edu)

**Abstract.** Near-surface air temperature (Ta) is a key variable in global climate studies. A global gridded dataset of daily maximum and minimum Ta (Tmax and Tmin) is particularly valuable and critically needed in the scientific and policy communities, but is
still not available. In this paper, we developed a global dataset of daily Tmax and Tmin dataset at 1-km resolution from 2003 to 2020 through the combined use of station-based ground Ta measurements and satellite observations (i.e., digital elevation model, and land surface temperature) via a state-of-the-art statistical method named Spatially Varying Coefficient Models with Sign Preservation (SVCM-SP). The root mean square errors of our estimates ranged from 1.20 to 2.44 for Tmax and 1.69 to 2.39 ℃ for Tmin. We found that the accuracies were affected primarily by land cover types, elevation ranges, and climate backgrounds. Our
dataset correctly represents the negative and positive relationships between Ta with elevation or land surface temperature; it captured spatial and temporal patterns of Ta realistically. This global 1-km gridded daily Tmax and Tmin dataset is the first of its kind and we expect it to be of great value to global studies such as urban heat island phenomenon, hydrological modeling, and epidemic forecasting. The data are available at Iowa State University's DataShare at https://doi.org/10.25380/iastate.c.6005185 (Zhang and Zhou, 2022).

**1 Introduction**

Near-surface air temperature (Ta) refers to the atmospheric temperature 1.5–2m above surfaces and it is an important variable for numerous applications, especially those pertinent to climate and environment change (Huang et al., 2019; Zhang et al., 2018), terrestrial hydrology and phenology (Lin et al., 2012; Ren et al., 2019), public health (Lan et al., 2010, 2022; Zhang et al., 2019), disease vectors propagating (Lowen et al., 2007; Petrova and Russell, 2018; Wu et al., 2020), and epidemic forecasting (Aggarwal
et al., 2012; Connor et al., 1998). Ta is related to land surface temperature (LST) but is inherently different from LST. Ta generally varies across space and time dramatically, due to the spatial heterogeneity and temporal dynamics of environmental factors such as solar radiation, wind speed, land cover, cloud cover, and vegetation phenology (Benali et al., 2012; Chen et al., 2015, 2021; Prihodko and Goward, 1997). At the global scale, a Ta dataset will be of limited or no use if not characterizing and capturing such fine spatial details and continuous temporal coverage. A high-/medium-resolution global Ta dataset at the daily interval is highly
desirable.

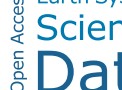

Many global or regional Ta datasets have been previously published (Chen et al., 2021; Crespi et al., 2021; Fang et al., 2021; Hersbach et al., 2018; Hooker et al., 2018; Kalnay et al., 1996; MacDonald et al., 2020; Meyer et al., 2019; Nashwan et al., 2019; Oyler et al., 2015; Thornton et al., 2021; Werner et al., 2019); however, these have either coarse spatiotemporal resolutions or only cover specific regions (Table S3). For example, some global Ta datasets have daily frequencies but at coarse spatial resolutions

(e.g., 0.05° or even coarser) (Hersbach et al., 2018; Hooker et al., 2018; Kalnay et al., 1996); other Ta datasets with medium spatial resolutions (~ 1-km) are only available for specific regions such as North America and mainland China (MacDonald et al., 2020; Oyler et al., 2015; Thornton et al., 2021; Chen et al., 2021). There are also several Ta datasets at even finer spatial resolutions but generated only for much smaller spatial regions (Crespi et al., 2021; Meyer et al., 2019; Nashwan et al., 2019; Werner et al., 2019). Despite the increasing need for a global daily Ta at finer resolutions (e.g., 1 km), such global products do not exist yet—a gap still

not filled yet.

Methodologically speaking, a range of techniques have been proposed and applied to generate Ta products; the majority of them rely on combining weather station data and gridded auxiliary datasets to simply make spatial interpolation or build empirical predictive models (Chen et al., 2015; Goward and Waring, 1994; Hengl et al., 2012; Hou et al., 2013; Hrisko et al., 2020; Li and Zha, 2019; Li et al., 2018; Nemani and Running, 1989; Rao et al., 2019; Shen et al., 2020; Shi et al., 2017; Sun et al., 2005; Yoo

et al., 2018; Zhu et al., 2013). Common spatial interpolation algorithms, such as Inverse Distance Weighting (IDW), Spline, and Kriging, are unlikely to be applicable at the global scale, for example, due to the relative sparsity of weather stations and the high spatial heterogeneity of Ta (Chai et al., 2011; Dodson and Marks, 1997; Li and Heap, 2011; Stahl et al., 2006). Model-based approaches are often a better choice to capture the true spatial variability of Ta; these methods are roughly divided into three groups. The first is the Temperature-Vegetation Index (TVX) method; it assumes that the temperature of a fully vegetated canopy

approximates near-surface Ta within the canopy such that Ta could be empirically estimated from the maximum normalized difference vegetation index (NDVI) point of the linear equation between NDVI and LST (Goward and Waring, 1994; Nemani and Running, 1989; Zhu et al., 2013). An apparent weakness of the TVX method is its large uncertainty or unsuitability for regions with low vegetation cover. The second group is the energy balance method, leveraging the explicit modeling of surface energy balance and the quantification of net radiation (e.g., the sum of sensible, soil, and latent heat fluxes) (Sun et al., 2005; Zhang et

al., 2015). Energy-based methods are physically based, requiring detailed characterization of surface biophysical conditions and thereby, making it difficult to implement for large areas, due to the lack of such detailed biophysical parameters.

Of the three groups, the third category is statistical methods that estimate Ta via statistical relationships empirically calibrated between Ta and other covariates. Common algorithms used include geographically weighted regression (GWR), Cubist, random forests, and deep learning, among others (Chen et al., 2015, 2021; Hooker et al., 2018; Li et al., 2018; Rao et al., 2019; Shen et al.,

2020; Yoo et al., 2018). Compared to physical-based methods, statistical methods have fewer restrictions on data requirements and better applicability for large spatial extents (Noi et al., 2017). However, direct applications of common statistical methods often fail to capture and preserve relationships between Ta and auxiliary covariates (e.g., an unrealistically positive relationship between Ta and elevation), thereby leading to large uncertainties or even wrong results. To overcome such drawbacks, we recently proposed a class of Spatially Varying Coefficient Models with Sign Preservation (SVCM-SP) (Kim et al., 2021; Zhang et al., 2022b), and

these can capture and preserve relationships between Ta and explanatory variables. The SVCM-SP algorithm was originally implemented for estimating Ta over mainland China, with significant improvement in terms of both accuracies and computational efficiency compared to alternative methods such as GWR (Zhang et al., 2022b). The potential of SVCM-SP as a routine algorithm to generate global Ta products is still untapped and addressed here.

Here we aim to generate a global dataset of daily maximum and minimum Ta (Tmax and Tmin) at 1 km across 50°S ~79°N

from 2003 to 2020 by integrating ground-based Ta measurements from weather stations and gridded satellite-observed covariates

Earth System
Science
Data

(i.e., DEM and LST). We employed our newly developed SVCM-SP algorithm that, for example, can preserve negative and positive relationships with elevation and LST, respectively. Our dataset is also aimed to improve upon existing published Ta datasets in both accuracies, spatial resolutions, and temporal coverage.

## 2 Study area and data

Land areas covered by our global dataset span from 50°S to 79°N. We divided the global lands roughly into five regions: North America, Latin America, Europe and Asia, Africa, and Australia/New Zealand. To encompass all the land areas resolved at 1-km resolution as well as cover all the possible weather stations, the boundaries of the five regions were irregular. There also exist some overlaps between the regions to reduce uncertainties of estimated Ta using the average in the overlapped areas. Our analysis considered three major data sources: ground-based Ta observations from weather stations, satellite-derived LSTs, and elevation.

In our algorithm, ground-based Ta is assumed to be statistically related to satellite LST and elevation. Details about each data source are further described below.

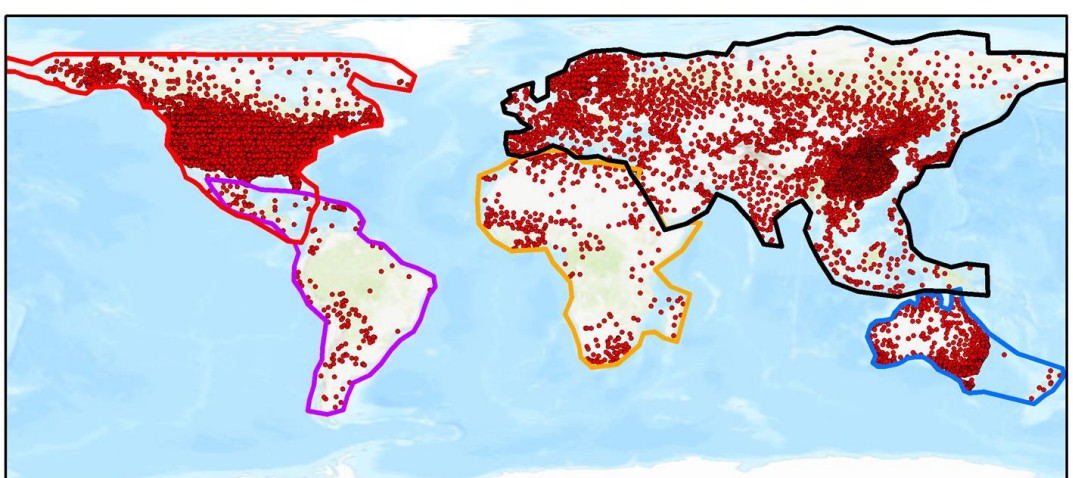

**Figure 1: Regions and locations of weather stations in this study. Red points are the locations of weather stations, polygons are the boundary of regions used in the SVCM-SP algorithm.**

Ground-based daily Ta measurements were compiled from a total of 103156 weather stations from 2003 to 2020. These are obtained from two climatology networks, the Global Historical Climatology Network daily (GHCNd) across the world and the China Meteorological Data Service Centre (CMDC) across mainland China. The LST dataset we used is a global seamless 1-km resolution daily (mid-daytime and mid-nighttime) LST dataset from 2003 to 2020; it was produced from MODIS 1-km daily LST product (MYD11A1 and MOD11A1) using a spatiotemporal gap-filling framework (Zhang et al., 2022a), and is available from

Iowa State University's DataShare platform (https://doi.org/10.25380/iastate.c.5078492). Both the mid-daytime and mid-nighttime LSTs were considered in our analysis. The DEM layer we chose is the SRTM30_PLUS product at 1-km resolution (Becker et al., 2009), which has been generated from the combination of the Shuttle Radar Topography Mission (SRTM30) topography (Hennig et al., 2001; Rosen, 2000) within a latitude of ±55 degrees, ICESat derived topography (Dimarzio et al., 2007) in Antarctica, and the GTOPO30 topography (Danielson and Gesch, 2011) in the Arctic. Besides, the Köppen-Geiger climate zones and MODIS land

cover data (MCD12Q1) were acquired to divide the world into zones for accuracy assessment (Sulla-Menashe and Friedl, 2018).

## 3 Methods



The core of our methodological framework is the SVCM-SP algorithm that correlates ground-based Ta with satellite LST and elevation. We applied the SVCM-SP algorithm to estimate Tmax and Tmin separately. To capture potential non-stationarity, the algorithm was trained for each day of the period 2003-2020 as well as for each of the five regions. Before applying the SVCM-SP algorithm, weather station Ta data were first pre-processed and filtered for quality control to ensure the high fidelity of reference Ta observations.

More specially, we first processed the weather station data in three ways. First, the locations of many weather stations in China, especially those located in complex terrains, are not accurately documented, geo-referenced only at the level of arc degrees and minutes in the metadata. Such location errors have to be corrected, and we manually corrected the locations of those weather stations located over complex terrains by searching the meteorological observation fields near the reported locations of weather stations with the help of high spatial resolution images from ArcGIS base map or google maps (Zhang et al., 2022b). Second, missing values are also prevalent, especially for those stations in Africa and Latin America. We filled these data gaps using a 5-day local moving window (Fig. S2). Third, the processed ground Ta data from the two steps were overlaid and matched with satellite LST and elevation to extract pairs of ground Ta and satellite covariates as inputs to the SVCM-SP algorithm.

The SVCM-SP algorithm seeks to build a spatially varying relationship between ground-based Ta with LST and elevation with sign preservation (Kim et al., 2021; Zhang et al., 2022b). A salient feature distinguishing it from conventional regression approaches is the spatially varying nature with constraints of estimated coefficients in the predictive relationship:

$$T_a(u_i, v_i) = \beta_0(u_i, v_i) + \beta_{elev}(u_i, v_i) \cdot Elevation(u_i, v_i) + \beta_{lst}(u_i, v_i) \cdot LST(u_i, v_i) + \varepsilon_i , \qquad (1)$$

where both the variables (e.g., Ta, Elevation, and LST) and the model parameters are functions of locations/coordinates $(u_i, v_i)$. More importantly, the two slope parameters (e.g., $\beta_{elev}$ and $\beta_{lst}$) are constrained to be negative and positive, respectively. $\varepsilon_i$ is the normal random error with mean zero and finite variance. These unknown parameters were estimated with a penalized bivariate spline method based on the triangulation technique under constraints. Details about the SVCM-SP algorithm are reported in Kim et al. (2021) and Zhang et al. (2022b). To estimate Ta across the globe, we applied the SVCM-SP algorithm to develop region-specific relationships for the five regions (Fig. 2). Also, two separate sets of equations were developed, one for Tmax using mid-daytime LST as the explanatory variable, and another for Tmin using mid-night LST as the explanatory variable. Accuracies of the estimated Ta were assessed using 10-fold cross-validation in these regions for each day. Metrics computed for accuracy assessment include root mean square error (RMSE) and mean absolute error (MAE).



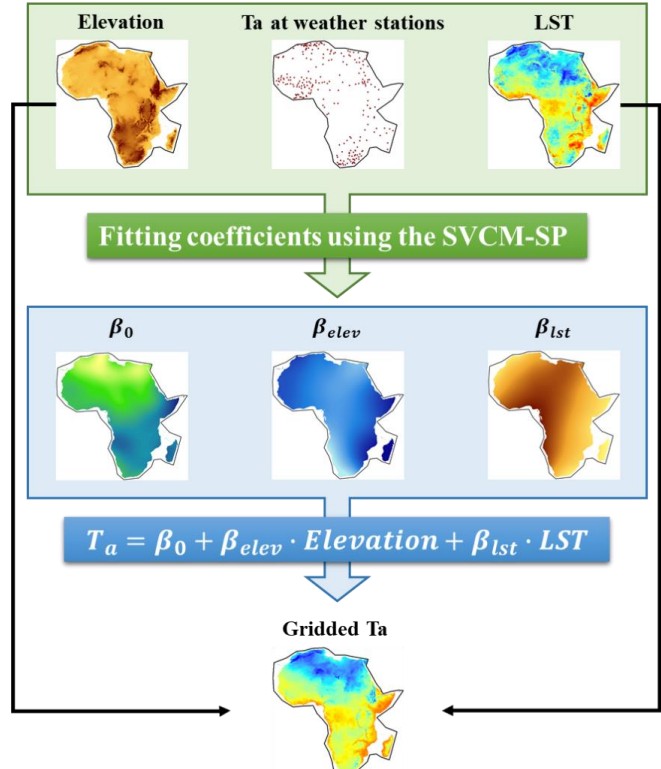

**Figure 2: Framework for implementing the SVCM-SP algorithm in a region (e.g., Africa). $\beta_0$, $\beta_{elev}$ and $\beta_{lst}$ are the intercept, coefficients of elevation, and LST, respectively.**


## 4 Results and discussion

### 4.1 Accuracy of the estimated Ta

The results of the 10-fold cross-validation indicate the accuracy of estimated Ta varies in different regions within a reasonable range (Table 1). The estimated RMSE and MAE ranged from 1.20 to 2.44 ℃ and 0.89 to 1.82 ℃, respectively. The highest accuracy

was obtained in Australia for Tmax, with the RMSE and MAE of 1.20 ℃ and 0.89 ℃, respectively. The lowest accuracy was obtained in North America for Tmax, with the RMSE and MAE of 2.44 ℃ and 1.82 ℃, respectively. Meanwhile, the indicators of accuracy have different values across years, but accuracies remained relatively consistent for each region (Tables S1-S2). The variations in accuracy may be caused by the differences in climate and topography in these regions (Hooker et al., 2018). For example, Australia is a continent with the gentlest undulations of terrains with about 87% of the land below 500 m.a.s.l. and is

dominated by hot arid desert and steppe climates, leading to the smallest spatial variations of Ta. However, other regions contain a variety of dominant climate types and geomorphic types, contributing to the large spatial variability observed in Ta.

**Table 1. Multi-year average accuracies for Tmax and Tmin in different regions from 2003 to 2020**

| Indicator | North America | | Latin America | | Europe&Asia | | Africa | | Australia | |
|---|---|---|---|---|---|---|---|---|---|---|
| | Tmax | Tmin | Tmax | Tmin | Tmax | Tmin | Tmax | Tmin | Tmax | Tmin |
| RMSE±SD (℃) | 2.44±0.40 | 2.39±0.39 | 1.94±0.42 | 1.97±0.30 | 1.80±0.19 | 1.75±0.26 | 2.22±0.55 | 2.21±0.40 | 1.20±0.26 | 1.69±0.26 |




| | | | | | | | | | |
|---|---|---|---|---|---|---|---|---|---|
| MAE±SD (ºC) | 1.82±0.30 | 1.78±0.29 | 1.45±0.29 | 1.47±0.22 | 1.29±0.15 | 1.28±0.20 | 1.62±0.37 | 1.68±0.29 | 0.89±0.18 | 1.28±0.21 |

Note: The selected testing stations were within 50 km² surrounding the training stations. SD represents the corresponding standard deviation.

The accuracy of estimated Ta varied in different land cover types, elevation ranges, and climate types (Tables 2-4). RMSE and

MAE for Tmax ranged from 2.06 to 2.56 ºC and 1.54 to 1.97 ºC, respectively, and these indicators for Tmin range from 1.84 to 2.83 ºC and 1.40 to 1.96 ºC, respectively. The model performs well for impervious surface (with the lowest RMSE), cropland, water, and wetland, whereas RMSE values were higher for the tundra and bare land, which was generally consistent with the findings of existing studies in mainland China (Chen et al., 2021; Shen et al., 2020; Zhang et al., 2022b). As shown in Table 3, RMSE and MAE values vary with elevation ranges but did not increase with the increase of elevation ranges, which is different

from existing findings (Chen et al., 2015; Rao et al., 2019). This is because we only used weather stations within the distance of 50 km from the training sites to evaluate the accuracy of estimated Ta in this study, which can mitigate the effects of sparse weather stations at high elevations on accuracy assessment, as reported in existing studies (Chen et al., 2015; Rao et al., 2019). RMSE and MAE values in equatorial climate zones are distinctly lower than those of other climate zones (Table 4), indicating the highest accuracies for both Tmax and Tmin, possibly due to Ta near the equator being generally warmer and less intra-annual variations

compared to other climate zones (Legates and Willmott, 1990).

**Table 2. Model performance for different land cover types in 2003-2020**

| Land cover type | Tmax | | | Tmin | | |
|---|---|---|---|---|---|---|
| | Records (%) | RMSE±SD (ºC) | MAE±SD (ºC) | Records (%) | RMSE±SD (ºC) | MAE±SD (ºC) |
| Cropland | 11.40 | 2.08±0.77 | 1.59±0.64 | 11.40 | 1.89±0.69 | 1.45±0.56 |
| Forest | 21.82 | 2.20±0.84 | 1.71±0.71 | 21.82 | 2.26±0.86 | 1.76±0.74 |
| Grassland | 38.01 | 2.29±0.86 | 1.76±0.72 | 38.00 | 2.29±0.84 | 1.77±0.73 |
| Shrubland | 1.58 | 2.13±0.94 | 1.66±0.77 | 1.57 | 2.36±1.07 | 1.89±0.97 |
| Wetland | 0.06 | 2.09±0.83 | 1.54±0.55 | 0.06 | 1.87±0.68 | 1.40±0.49 |
| Water | 1.60 | 2.09±0.75 | 1.61±0.60 | 1.59 | 2.07±0.86 | 1.61±0.71 |
| Tundra | 0.77 | 2.56±1.40 | 1.97±1.27 | 0.77 | 2.83±1.30 | 2.17±1.12 |
| Impervious surface | 21.49 | 2.06±0.75 | 1.58±0.62 | 21.53 | 1.84±0.62 | 1.41±0.50 |
| Bare land | 3.23 | 2.22±0.84 | 1.71±0.70 | 3.21 | 2.46±0.95 | 1.96±0.84 |
| Snow/Ice | 0.05 | 2.18±0.67 | 1.71±0.59 | 0.05 | 2.46±0.75 | 1.91±0.61 |

Note: SD represents the corresponding standard deviation.

**Table 3. Model performance for different elevation ranges in 2003-2020**

| Elevation (m) | Tmax | | | Tmin | | |
|---|---|---|---|---|---|---|
| | Records (%) | RMSE±SD (ºC) | MAE±SD (ºC) | Records (%) | RMSE±SD (ºC) | MAE±SD (ºC) |
| < 1000 | 75.36 | 2.13±0.80 | 1.63±0.66 | 75.37 | 2.01±0.72 | 1.54±0.59 |
| 1000-2000 | 16.84 | 2.44±0.90 | 1.89±0.76 | 16.84 | 2.51±0.99 | 1.99±0.87 |



| | | | | | |
|---|---|---|---|---|---|
| 2000-3000 | 6.76 | 2.24±0.88 | 1.75±0.76 | 6.76 | 2.71±0.98 | 2.18±0.87 |
| 3000-4000 | 1.01 | 2.11±1.04 | 1.65±0.80 | 1.00 | 2.44±0.87 | 1.95±0.75 |
| > 4000 | 0.03 | 2.69±1.05 | 2.28±1.00 | 0.03 | 2.48±0.57 | 1.90±0.43 |

Note: SD represents the corresponding standard deviation.

**Table 4. Model performance for different climate types in 2003-2020**

| Climate type | Tmax | | | Tmin | | |
|---|---|---|---|---|---|---|
| | Records (%) | RMSE±SD (ºC) | MAE±SD (ºC) | Records (%) | RMSE±SD (ºC) | MAE±SD (ºC) |
| Equatorial | 1.36 | 1.54±0.67 | 1.24±0.61 | 1.39 | 1.49±0.73 | 1.20±0.69 |
| Arid | 16.72 | 2.34±1.00 | 1.80±0.85 | 16.69 | 2.33±0.91 | 1.83±0.79 |
| Warm temperate | 46.28 | 2.14±0.76 | 1.65±0.64 | 46.29 | 1.94±0.76 | 1.50±0.64 |
| Snow | 34.99 | 2.21±0.82 | 1.70±0.68 | 34.98 | 2.36±0.80 | 1.82±0.68 |
| Polar | 0.65 | 2.32±1.12 | 1.80±0.94 | 0.65 | 2.35±0.93 | 1.81±0.76 |

Note: SD represents the corresponding standard deviation.

Spatial distributions of RMSE illustrate that most of the climate zones show reasonable accuracies (RMSE < 3.0 ºC) for Tmax

and Tmin (Fig. 3). The lower-accuracy climate zones (RMSE > 3.0 ºC) mainly occur where there are low station densities (Fig. S6), which is consistent with the finding of decreasing accuracy with the increase of station density (Shen et al., 2020). Meanwhile, these lower-accuracy climate zones are generally located at the boundary of regions where some directions have no weather stations.



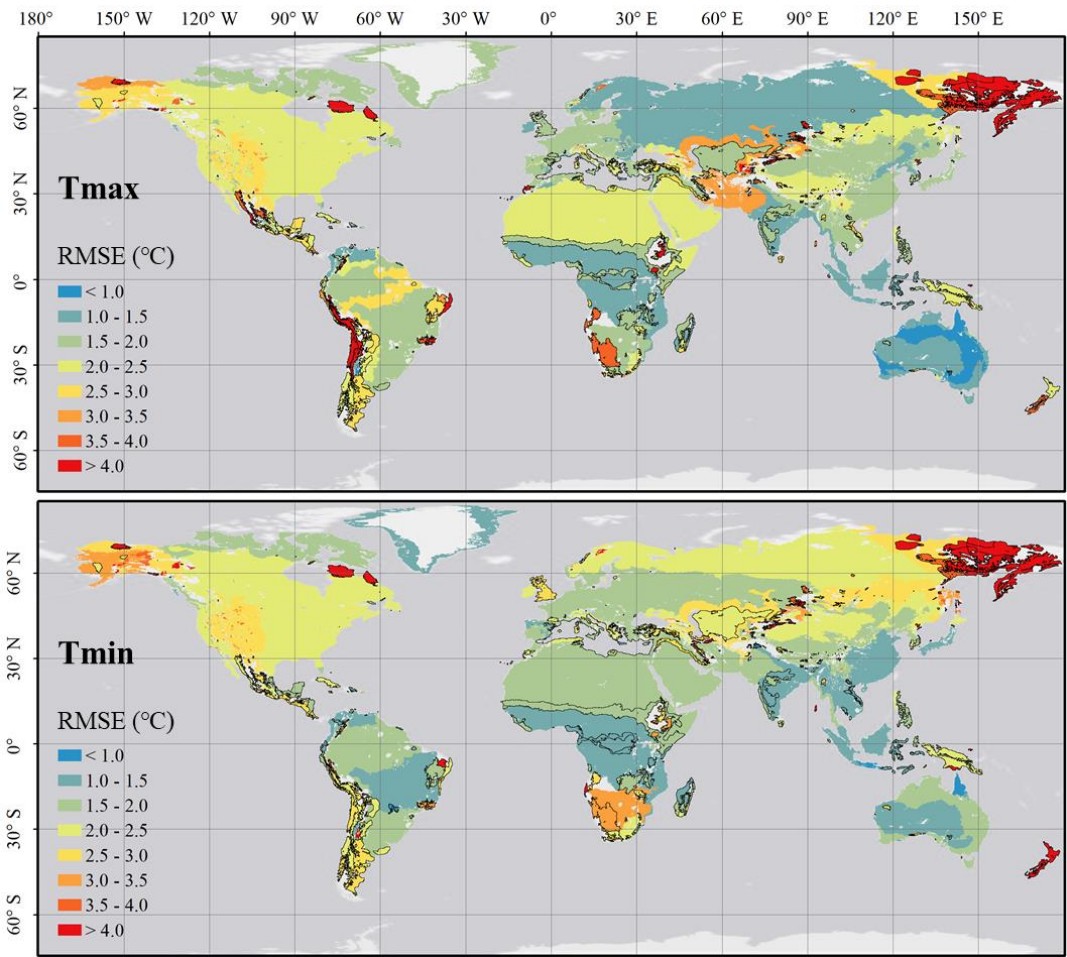

**Figure 3: Spatial pattern of accuracy in estimated Ta of different climate zones in 2003-2020. Climate zones with black boundaries do**
**not contain available weather stations within 50 km of the training stations for accuracy assessment, i.e., the validation data come from**
**weather stations further than 50 km away from the training stations.**

The RMSE values generally show distinctly seasonal patterns in the five regions within reasonable ranges (Fig. 4). Taking the
year 2010 as an example, RMSEs in Summer (June, July, and August) are generally lower than those in Winter (December, January,
and February) in North America, Europe and Asia regions (Fig. 4), possibly due to plant phenology which leads to a closer
relationship between Ta and LST in the Summer than that of the Winter (Benali et al., 2012; Cai et al., 2017; Lin et al., 2012). This
seasonal variation is less obvious in Africa and Latin America, possibly due to weaker correlations between plant phenology and
air temperature in the two regions, which are located across the equator (Adole et al., 2019; Sakai and Kitajima, 2019). Specifically,
the Australia region, which locates in the Southern Hemisphere, shows higher RMSEs in Summer (December, January, and
February) than that in Winter (June, July, and August) for Tmax. This may be caused by more homogeneous spatial variations of
Tmax in Winter than that in Summer in the Australian region.

**Figure 4: Temporal patterns of accuracies in estimated Ta in different regions in the year 2010.**

## 4.2 Spatial and temporal patterns of Ta

The estimated Ta shows significant spatial variations at the global scale (Fig. 5). Taking the estimated Ta in one July day as an example, both Tmax and Tmin decrease from about 30º N to the North and South Poles (Fig. 5). Meanwhile, lower Ta values also occur at higher elevation regions such as the Tibetan Plateau in the center of Asia and the Andes Mountains in the west of South America. Therefore, the characteristics of Ta change with latitude and elevation (i.e., the trend of lower Ta in higher latitude/elevation areas), which is consistent with the existing studies (Chen et al., 2015; Zhang et al., 2022b). The highest Ta values occur in northern Africa and the Arabian Peninsula, as these regions are mainly covered by the Gobi deserts.



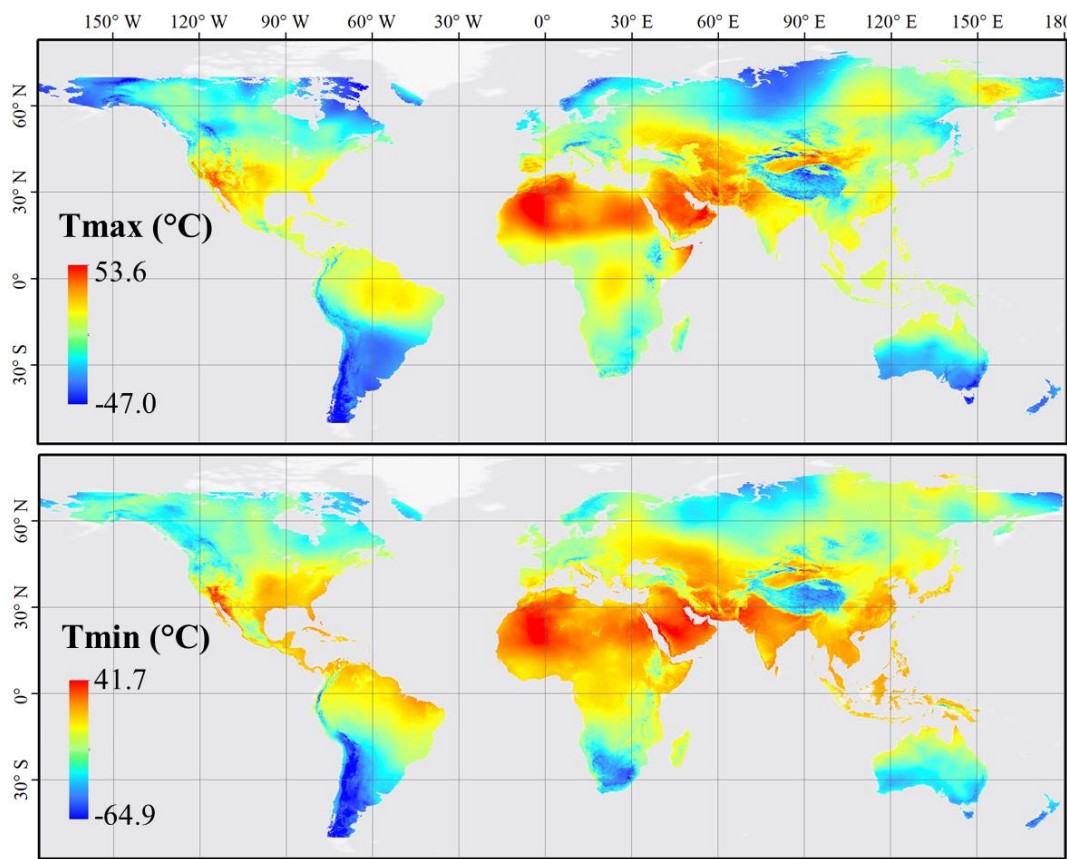


**Figure 5: Spatial pattern of estimated Ta at the global scale in an example day of 200 in 2010.**

The spatial patterns of estimated Ta in selected cities with clear weather around the world illustrate that the urban heat island (UHI) phenomenon (i.e., the higher temperature in urban than in the surrounding rural areas) has been well captured at the city scale (Fig. 6). On an example day of July in 2010, the estimated Ta in these cities shows an obvious UHI phenomenon, which is

reasonable with the transition from urban centers to surrounding rural areas. The estimated Ta in Changsha, China, shows several hotspots because some nearby cities (such as Xiangtan and Zhuzhou) have also been included in the buffer of Changsha, indicating the effectiveness of the estimated Ta for presenting UHI in small urban areas. Specifically, as a coastal city, estimated Ta in Melbourne, Australia, shows decreasing trends from the coast, and the UHI phenomenon is not obvious in surrounding small cities. This is because there is also an increasing trend of elevation from the coast in Melbourne, leading to the mixed spatial patterns of

Ta due to the UHI phenomenon and elevation changes.

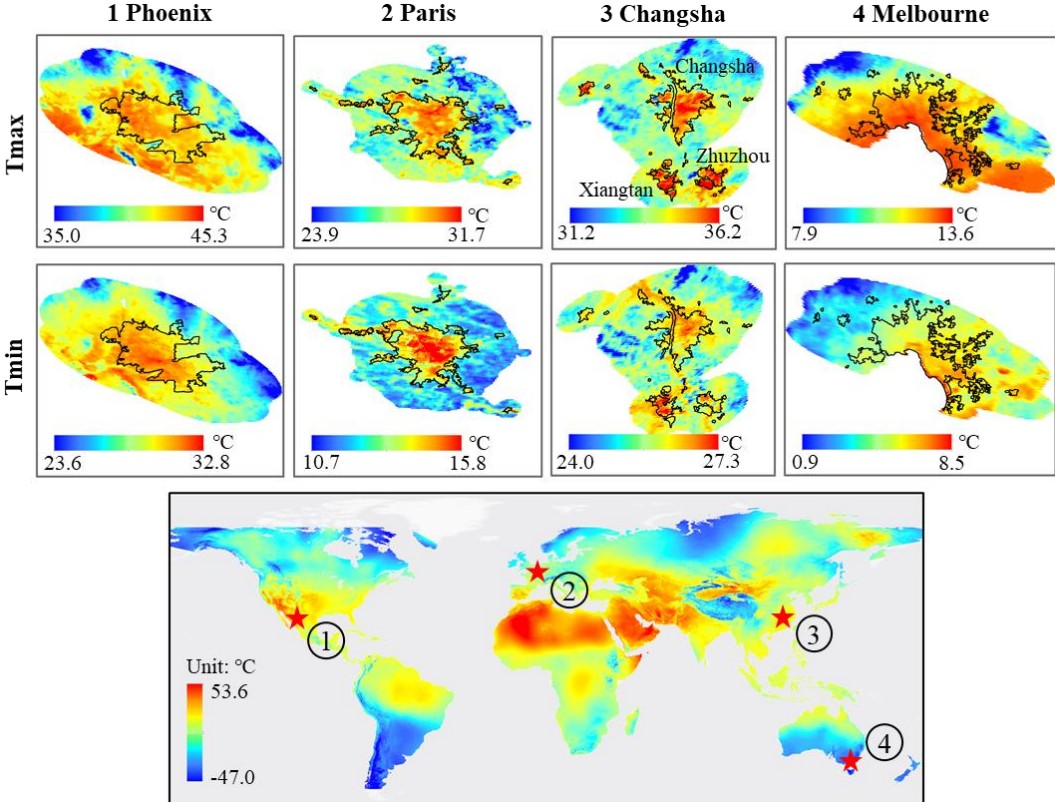

**Figure 6: Spatial pattern of estimated Ta in five representative cities on day 200 of the year 2010. A city shape includes the urban extracted by using nighttime light data (Zhou et al., 2018) and its surrounding buffer of equal size. Black color solid lines are the boundary of urban regions extracted by using global artificial impervious area data with 30 m spatial resolution (Li et al., 2020).**

The comparison of the temporal pattern between estimated Ta and ground-based measurements from an example of weather stations in a mega-city (Fig. 7) illustrates that the SVCM-SP algorithm can effectively (RMSE of 1.25°C and 1.53°C, respectively, for Tmax and Tmin) estimate Ta for the entire period. As shown in Fig. 7, the estimated Ta based on 10-fold cross-validation and Ta observations from the weather station in Beijing, China, show similar temporal patterns and very close values for both Tmax and Tmin in 2010. For both clear weather (days 28 and 130 in Fig. 7) and overcast weather (days 219 and 293 in Fig. 7) (Zhang et al., 2022a), the gridded Ta can illustrate the UHI phenomenon. An existing study has found that the estimated Ta in urban areas was more accurate than those of other regions (Zhang et al., 2022b), specifically suggesting its great value for urban applications.



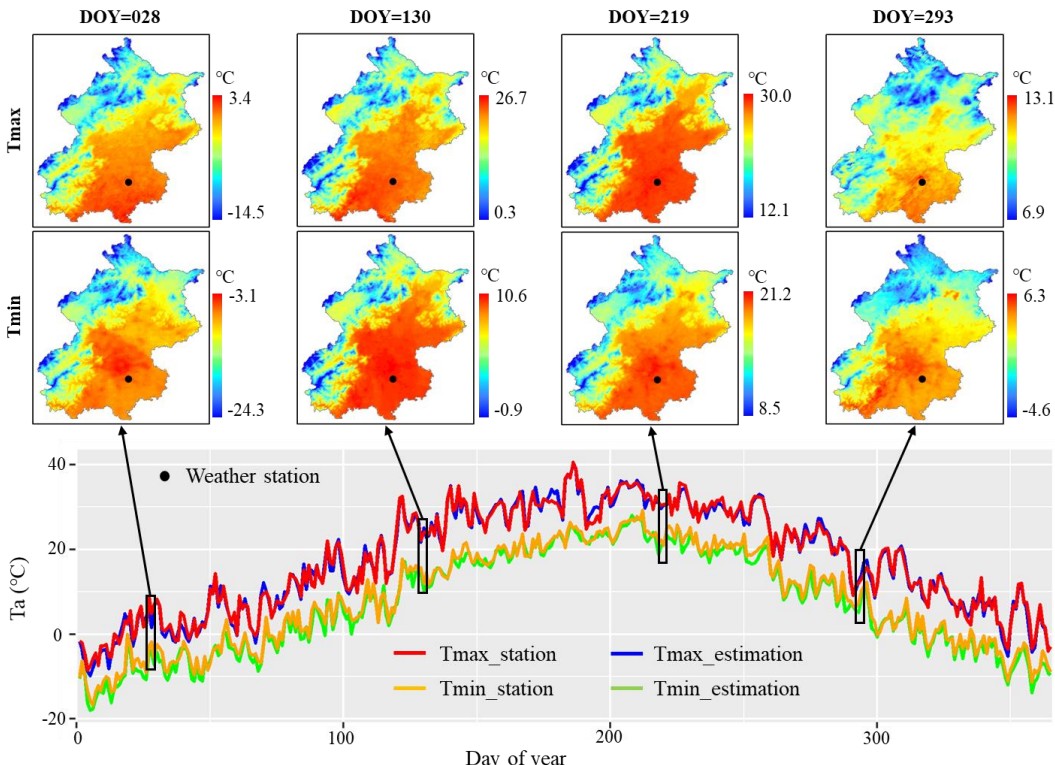

**Figure 7: Temporal pattern of estimated and observed Ta in the weather station of Beijing (black point) in the year 2010. The black rectangles are example days showing maps of estimated data in Beijing.**

### 4.3 Comparison with existing Ta datasets

The gridded Ta data in this study have advantages regarding spatiotemporal resolutions (i.e., 1-km daily maximum and minimum) or coverage (i.e., global) (Table S3). In the existing Ta datasets, global Ta data usually have relatively low spatial resolutions. For example, Ta from ERA5 and NCEP/NCAR reanalysis datasets have a spatial resolution of 0.25° and 2.5°, respectively, although their coverage time (1979 to date, and 1948 to date, respectively) and temporal frequencies are satisfactory (hourly and 4-times per day, respectively). Besides, Hooker et al.(2018) generated global Ta with 0.05° spatial resolution on a monthly scale from 2003 to 2016. Ta datasets that have improved spatial resolutions are usually available on a continental/national scale (Chen et al., 2021; Fang et al., 2021; MacDonald et al., 2020; Oyler et al., 2015; Thornton et al., 2021) and can reach 1-km spatial resolution and daily frequency. Crespi et al.(2021) created the Ta dataset with 250 m spatial resolution daily frequency from 1980 to 2018, but it is only available in North-eastern Italy. The Ta data in this study have a spatial resolution of 1 km and include daily Tmax and Tmin with global coverage (50°S ~79°N) from 2003 to 2020, which have higher spatiotemporal resolutions or spatial coverage than other existing published Ta datasets.

The gridded Ta in this study can effectively capture the spatial variation of Ta under clear physical meanings (i.e., negative and positive relationship with elevation and LST, respectively), which is not always true in other gridded Ta datasets. The existing Ta datasets were created using regression methods such as PRISM (Crespi et al., 2021), thin plate smoothing spline models (MacDonald et al., 2020; Werner et al., 2019), and GWR (Hooker et al., 2018), and machine learning methods such as random forest (Chen et al., 2021; Meyer et al., 2019), which had no explicit constraints on the relationship between Ta with elevation and/or LST. The normal temperature lapse rates were considered using a parameter named vertical temperature gradient for





estimating Ta in Daymet, but the temperature lapse rates were limited to at most a 12 °C decrease and 1 °C increase in temperature per 1000 m elevation increase (Thornton et al., 2021), which is not a fully negative relationship between Ta and elevation. Scholars

have tried to build vertical lapse models to estimate gridded Ta according to Adiabatic Lapse Rate (ALR) (Dodson and Marks, 1997; Rhee and Im, 2014; Zhu et al., 2017), but the universality of these models is limited because it is difficult to accurately capture ALR due to its dramatical changes across space. In this study, the coefficients of elevation and LST were constrained as negative and positive, respectively, to restrict the corresponding relationship between Ta with elevation and LST (Fig. S7), which is more reasonable than existing Ta datasets.

The accuracy of the resulting gridded Ta from this study is comparable to several other reported gridded Ta datasets (e.g., Chen et al., 2021; Oyler et al., 2015; Thornton et al., 2021). Among them, the 1-km daily Ta from Daymet (Thornton et al., 2021) reaches MAE of 1.52 and 1.78 °C for Tmax and Tmin, respectively, and the 30-arcsec (~800 m) daily Ta from TopoWx (Oyler et al., 2015) reaches that of 1.03 and 1.06 °C, while in this study, the average MAE is 1.82 and 1.78 °C in North America. However, Daymet failed to capture the UHI phenomenon due to the spatial interpolation of Ta being implemented based on only elevation (Menne

et al., 2012) and did not consider the impact of biophysical and socioeconomic factors on spatial variations of Ta (Li et al., 2018). The estimated Ta from TopoWx can display the UHI phenomenon but tend to overestimate the impact of topographical features and show fewer temporal variations of the spatial pattern of Ta within a month than that in this study, as 10-year average of monthly LSTs were used as a covariate in TopoWx (Li et al., 2018; Oyler et al., 2015) instead of daily LST data in this study. The 1-km daily average Ta data by Chen et al.(2021) reaches RMSE of 1.615 to 1.957 K using leave-location-out cross-validation in mainland

China, while the average RMSE of estimated Tmax and Tmin is 1.80 and 1.75 °C, respectively, in Europe and Asia. While the accuracy of Ta obtained in this study is comparable to the other large-scale Ta datasets, our dataset is produced at the global scale using consistent modeling and assessment approaches.

There are some limitations in the SVCM-SP algorithm used in this study for creating the gridded Ta dataset, and future work can focus on improving the accuracy of the estimated Ta with an improved SVCM-SP algorithm. First, we only considered the

linear relationship between Ta and covariates. However, nonlinear relationships may exist between Ta with elevation and LST when other factors, such as winds, clouds, snow, and land cover types, have non-negligible impacts on Ta (Cai et al., 2017; Good, 2016). Second, we only used two covariates in the SVCM-SP algorithm, and the estimated Ta might be highly similar to the spatial patterns of LST due to the possible heavy dependence of Ta on the LST data. An applicable solution is using additional covariates (e.g., other surface characters such as GLAS-derived canopy height and vegetation parameters) in the SVCM-SP algorithm with

the linear or nonlinear relationships with Ta may further improve the model performance.

**5 Data availability**

Data described in this paper can be accessed at Iowa State University's DataShare at https://doi.org/10.25380/iastate.c.6005185 (Zhang and Zhou, 2022). The dataset contains 36 sub-datasets (one for Tmax and Tmin of each year from 2003 to 2020). Each sub dataset contains Tmax or Tmin of a specific year (2003–2020) in five regions (i.e., North America, Latin America, Europe and

Asia, Africa, and Australia (and New Zealand)) and is organized by day of the year. The data are in GeoTIFF with the georeferenced information embedded. Each file keeps the MODIS ellipse sinusoidal projection with a spatial resolution of 1 km. The unit of LST in GeoTIFF is 0.1 degrees Celsius (°C), and the naming rule can be found in the file "README.pdf".

**6 Conclusions**

We generated a global (50°S ~79°N) 1-km daily maximum and minimum Ta (i.e., Tmax and Tmin) dataset from 2003 to 2020 based on ground-based Ta measurements from weather stations and gap-filled LST dataset using the Spatially Varying Coefficient Models with Sign Preservation (SVCM-SP) algorithm. The dataset showed acceptable accuracies based on the 10-fold cross-validation for five regions of the globe, compared to existing Ta datasets. The RMSEs of estimated Tmax and Tmin ranged from 1.20 to 2.44 and 1.69 to 2.39 ℃, respectively. The estimated Ta was affected by land cover types, elevation ranges, and climate types, with varying accuracies but within reasonable ranges. Our gridded Ta dataset effectively captured the spatial variation of Ta

under clear physical meanings (i.e., negative and positive relationship with elevation and LST, respectively), which is not always true in other gridded Ta datasets. The new dataset is unique in terms of spatiotemporal resolutions (i.e., 1-km daily maximum and minimum), global coverage, and temporal span and should be useful for a wide range of applications such as urban heat island phenomenon, hydrological modeling, and epidemic forecasting. However, the gridded Ta dataset may be limited by the performance of the SVCM-SP algorithm because only linear relationships between Ta with elevation and LST were used in the

model. Therefore, future work can focus on improving the performance of the SVCM-SP algorithm by using more explanatory variables under proper considerations of the relationship with Ta.

**Supplement.**

**Author contributions.**

YZ designed the research, TZ implemented the research and wrote the original manuscript, and YZ and ZZ supervised the research. All co-authors revised the manuscript and contributed to the writing.

**Competing interests.**

At least one of the (co-)authors is a member of the editorial board of *Earth System Science Data*. The peer-review process was guided by an independent editor, and the authors also have no other competing interests to declare.

**Acknowledgments.**

This research was supported by the College of Liberal Arts and Science (LAS) Dean's Emerging Faculty Leaders award at the Iowa State University and the National Science Foundation (2041859, 1803920, and 2203207).

**Financial support.**

**Review statement.**

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
