# Peer review of "A global dataset of daily maximum and minimum near-surface air temperature at 1-km resolution over land (2003-2020)"

_Earth System Science Data, 2022_

## Author Comment (AC1)

**Reviewer #1**

**Comment #1-1**: Satellite remote sensing and ground-based stations play their unique advantages in global meteorological parameter retrieval, and near-surface temperature is of great significance to global and local climate. In this study, satellite and ground-based data are used to retrieve near-surface maximum and minimum temperatures in almost global land regions with 1-km resolution. Overall, this manuscript is clear and well written and presents interesting research, however I found there was a lack of details necessary to fully understand the methods. Some concerns are needed to address, below are my major comments. (Major revision)

**Response:** Thank you very much for your suggestions. We have provided more details to clarify our method. We first clarified the similarities and differences between current study and the study by Zhang et al. (2022b). We then explained the time difference between LST and Ta, the suitability of 10-fold cross-validation, improved figures, and also added scatter plots for validation of the retrieval results. Below please find our responses to your comments in detail.

**Comment #1-2**: Line 16: I think "ground station-based Ta" is better than "station-based ground Ta".
**Response:** Done!

**Comment #1-3**: The descriptionof the "global dataset" is not rigorous, as the authors have only achieved retrieval of near-surface temperatures for most of the global land region.
**Response:** Thank you for your suggestion. We really appreciate your point. We keep 'global dataset' in the title to follow the conventions in the community. For example, the famous CRU dataset (https://www.ipcc-data.org/observ/clim/cru_climatologies.html) is one of the best global climate datasets although it is land-only. But we changed the title to 'A global dataset of daily maximum and minimum near-surface air temperature at 1-km resolution over land (2003-2020)' in the revised manuscript as suggested.

**Comment #1-4**: Lines 82-83: Different regional retrieval models have certain differences. Two different Tmaxs and Tmins will be obtained in the overlapping region. How does the author calculate the final Tmax and Tmin results?
Response: Thank you for your question. There is a significant overlapping region between North America and Latin America in our dataset. We agreed that different Tmaxs and Tmins can be obtained in the overlapping region. In the revised dataset, we removed the largely overlapping region between Latin America and North America to help users to use the dataset.

[Figure]

*Figure 1: Regions and locations of weather stations in this study. Red points are the locations of weather stations, polygons are the boundary of regions used in the SVCM-SP algorithm. Specifically, polygons of red, purple, orange, blue, and black represent the boundaries of North America, South America, Africa, Australia, and Europe & Asia, respectively.*

**Comment #1-5**: The latitude and longitude grid should be added in Figure 1.

Response: We have improved the figure in the revised manuscript as suggested.

**Comment #1-6**: Please explain in detail the similarities and differences with the study by Zhang et al. (2022b), and whether there are other differences besides the study area.

Response: Thank you for your question. The study by Zhang et al. (2022b) focuses on the development of the SVCM-SP algorithm for estimating gridded Ta, taking mainland China as an example. It contains many details on the SVCM-SP algorithm and was systematically compared with the geographically weighted regression (GWR) for the novelty of the method. As ESSD is interested in the publication of articles on original research data (sets), this paper focused on the characteristics (e.g., accuracy, spatial and temporal patterns) of the original gridded Ta dataset using the developed and validated SVCM-SP algorithm from Zhang et al. (2022b). Research papers are not within the scope of ESSD. It contains meticulous model calibration and accuracy assessment on the data product in different continents, land cover types, elevation ranges, and climate zones. It also includes several examples on the spatial and temporal distribution of Ta data. Besides, the advantages of the gridded Ta were shown by comparing with existing gridded Ta datasets. We added descriptions in the revised manuscript and removed replicated texts between the two studies.

*"Zhang et al. (2022b) successfully estimated and validated gridded Ta using the SVCM-SP algorithm and demonstrated its novelty through the comparison with the geographically weighted regression (GWR) model, while in this study, we developed the global product of gridded Ta, performed extensive model calibration and accuracy assessment at the global scale, and provided details on accuracy, spatial and temporal patterns of the global gridded Ta." (lines 76-79)*

**Comment #1-7**: Lines 107-114: The ground-based stations usually have a high temporal resolution, and the seamless global surface temperature data used by the author comes from the MODIS sensor, which has a limited number of transits. When the training sample library is constructed, the author only describes the spatial matching process and ignores the temporal matching, please give a detailed description.

**Response:** Thank you for your suggestion. Mid-daytime and mid-nighttime LSTs were used to develop their relationship with air temperature to interpolate station Tmax and Tmin, respectively. We agree that there may be time difference between Ta and LST used in the algorithm. The key information about LST in our algorithm is its spatial variations. Within the small difference in time between LST and Ta, there will not be significant change in the spatial variations of LST. Therefore, the impact of time difference between LST and Tmax/Tmin on the accuracy of the estimated Ta is minor. We proved it by adding shifts to LST for estimating Ta in North America in an example (Fig. S8). We added discussion in the revised manuscript.

*"Specifically, mid-daytime and mid-nighttime LSTs were used to develop their relationship with air temperature to interpolate station Tmax and Tmin, respectively. The actual time of Tmax and Tmin may be slightly different from mid-daytime and mid-nighttime of LST. Within the small difference in time between*

*LST and Tmax/Tmin, there will not be significant change in the spatial variations of LST. Therefore, the impact of time difference between LST and Tmax/Tmin on the accuracy of the estimated Ta is minor as shown by shifting LST for time difference (Fig. S8)." (lines 118-123)*

[Figure]

*Figure S8: Residuals of estimated Ta based on the 10-fold cross-validation by shifting LSTs from -5 to 5 ℃ at a step of 0.5 ℃ in North America in the day 200 of 2010. Each box is based on the residual between observed and estimated Ta in the stations. Each point represents the residual in a specific station.*

**Comment #1-8**: Lines 124-125: Does the authors mean that the ground-based measurement result corresponding to mid-daytime is the Tmax of the site, and the ground-based measurement result corresponding to mid-nighttime is the Tmin of the site? If yes, I don't think it's reasonable, especially for the Tmin.

Response: Thank you for your question and comment. As we explained in the response to your comment #1-7, we did not equate mid-daytime LST to Tmax or mid-nighttime LST to Tmax. Instead, our models are empirical/correlative in nature, and mid-daytime (or mid-nighttime) LST is correlated strongly with the true Tmax (or Tmin), which is the theoretical basis for our method as well as for all the existing methods in the literature. We agree that there may be time difference between Ta and LST used in the algorithm. But the impact of such differences in time on the interpolated Tmax/Tmin is minor. Please see details in our previous response.

**Comment #1-9**: Due to the spatial correlation of the near-surface air temperature at the different stations and the temporal correlation between the training data at same station, the 10-fold cross-validation verification cannot truly reflect the accuracy of the model. The author needs to give independent verification results, such as the first 15 days of each month as training Set, the data of the last 5 days is used as the validation set or test set, or the data from 2003-2018 is used as the training set, and the data in 2020 is used as the validation or test set, or 80% of the site data is used as the training set, and the data from the remaining 20% sites is used as the validation or test set.

**Response:** Thank you for your suggestion. Our cross-validation is a very rigorous one and the uncertainty estimates are based on independent data (not used for the model training). More importantly, the RMSE represents a conservative estimate of the true uncertainties of our data because when producing the final results, we use all available data, more than those in the 10-fold cross-validation. In order to evaluate the model performance for estimating gridded Ta, we used the widely used 10-fold cross-validation for each day in each region. That is, we equally and randomly divided the valid records into 10 groups. Nine groups were used as training set and the rest one group was used as testing set. This approach was implemented for 10 times until all the groups had been used as testing set. Each test of the 10-fold cross-validation can obtain a RMSE and the average RMSE of the 10 tests was used as the final RMSE. Therefore, the accuracy assessment of the 10-fold cross-validation was implemented based on independent validation data and can provide a reliable evaluation of the accuracy. We compared daily RMSEs based on 10-fold cross-validation and validation with 30% randomly selected testing data (Fig. R1). We found that the two validation methods show similar results in accuracy. The distributions of RMSEs using the 10-fold cross-validation method is more concentrated with higher maximum densities than those of the 30%-random-validation-based method, especially in Africa and South America due to the low number of validation records, indicating the superiority of the 10-fold cross-validation method in stability. We have added relevant descriptions in the revised manuscript.

*"The model performance for estimating gridded Ta was assessed based on root mean square error (RMSE) and mean absolute error (MAE) using the 10-fold cross-validation in these regions in each day. Taking the RMSE as an example, a RMSE was generated in each test of the 10-fold cross-validation and all RMSEs from the 10 tests were averaged as the final RMSE in a specific day in a specific region. This accuracy assessment using the 10-fold cross-validation was implemented based on independent validation data and can provide a reliable evaluation of the accuracy." (lines 134-139)*

*"Specifically, this accuracy assessment represents conservative estimates of the uncertainties of our data because when producing the final results, we used all the available data, more than those in the 10-fold cross-validation." (lines 141-143)*

[Figure]

Figure R1: Density of daily RMSEs from 30%-random-validation (red) and mean RMSEs from 10-fold cross-validation (blue) in five regions in year 2010.

**Comment #1-10**: What does the Y-axis of Figure S1 represent? Is it the number of valid observation samples or the number of stations that only include 1 valid observation sample? Also, I did not find a related description of Figure S1 in the manuscript.

**Response:** Thank you for your questions. The Y-axis of Figure S1 represents the number of stations with valid records in each day for Tmax (or Tmin). Using the daily number of valid records, we drew a boxplot in each year. We have clarified the descriptions (Figures S1 and S2) in the supplement.

*"Figure S1: Number of valid records (Y-axis) in five regions. Each box is based on the daily number of valid records in a specific year. Each point represents the number of valid records in a specific day."*

*"Figure S3: Number of valid records (Y-axis) after filling missing values in Africa and Latin America. Each box is based on the daily number of valid records in a specific year. Each point represents the number of valid records in a specific day."*

*"Second, there are missing values, especially in stations in Africa and Latin America (Fig. S1). We filled these data gaps using a 5-day local moving window (Fig. S2). Accordingly, the number of records largely increased (Figs. S3 and S4) with reasonable error ranges (Fig. S5)." (lines 114-117)*

**Comment #1-11**: Comparing Figure 1 and Figure 3 confuses me. In my view, the author has constructed Ta estimation models in 5 study regions, but does not include Greenland (Figure 1). I have three questions. First of all, why can the Ta retrieval in the Greenland region be achieved? The parameters of the retrieval model are the same as in which regions Ta retrieval model? In addition, the standardized regression coefficient for the Greenland region in Figure S7 is also absent. Second, can the constructed retrieval model be used for the retrieval of Ta in the oceanic region? Third, what do the non-color-filled regions on land in Figure 3 (such as the Amazon region and south-central Africa) mean?

**Response:** Thank you for your questions. (1) Our dataset covers a small portion of Greenland which is constrained by the extent of the global seamless 1 km daily LST dataset. (2) The dataset focuses on the land areas as indicated by the revised title. (3) The non-color-filled regions on land in Figure 3 (Figure 4 in the revised manuscript) are areas without reliable evaluations due to the lack of weather stations. We clarified these in the revised manuscript. As suggested, we improved relevant figures (Figs. 4 and S6) in the revised manuscript and supplement by only showing the regions covered by our dataset.

*"Specifically, our dataset covers a small portion of Greenland which is constrained by the extent of the global seamless 1-km daily LST dataset." (lines 102-103)*

[Figure]

*Figure 4: Accuracy of estimated Ta in climate zones in 2003-2020. Climate zones with black boundaries are areas with low densities of weather stations (i.e., distances between training and validation sites are larger than 50 km). The white regions on land are areas without reliable evaluations due to the lack of weather stations. (lines 191-193)*

[Figure]

*Figure S6: Station density in climate zones in 2003-2020. Climate zones with black boundaries are areas with low densities of weather stations (i.e., distances between training and validation sites are larger than 50 km). The white regions on land are areas without reliable evaluations due to the lack of weather stations.*

**Comment #1-12**: It is suggested to modify the color bar range of Tmax and Tmin in Figure 5 to be the same

**Response:** We improved the figure as suggested.

**Comment #1-13**: I think that the validation of the results in this manuscript needs to be expanded further and needs to add scatter plots for the validation of the retrieval results, such as density scatter plots instead of just calculating RMSE and MAE.

**Response:** As suggested, we have added scatter plots in 2010 for validation of the retrieval results.

*"The results of the 10-fold cross-validation indicate the accuracy of estimated Ta varies across regions within a reasonable range (Fig. 3 and Table 1). The estimated and observed Ta in different regions scattered along the 1:1 line with the RMSE ranging from 1.17 to 2.38℃ and 1.59 to 2.34℃, respectively, for Tmax and Tmin in 2010 (Fig. 3)." (lines 149-151)*

[Figure]

*"Figure 3: Scatter plots between estimated and observed Ta in five regions in year 2010. Each point represents the estimated and observed Ta (Tmax or Tmin) in a specific day in a weather station. The color of points represents the density, in which red and blue points represent the high and low densities, respectively. The red line is the regression line and the black line is the 1:1 line." (lines 163-165)*

---

## Author Comment (AC2)

**Reviewer #2**

Tmax and Tmin are both very important of ecosystem. High quality of long period and high resolution of air temperature products estimated from satellites are still lacking. This paper provided a set of such grided data with elaborated accuracy evaluations. However, there are some key questions needed to answer.

**Response:** Thank you for your comments and suggestions. Below please find our responses to your comments in detail.

**Comment #2-1**: I suggest you change the title because you olny produced Tmax and Tmin. 'Daily' means too much. Besides,'global' is more than what you did.

**Response:** Thank you for your suggestion. We have changed the title as 'A global dataset of daily maximum and minimum near-surface air temperature at 1-km resolution over land (2003-2020)'.

**Comment #2-2**: Line 77-78: I don't think you improved accuracies of spatial resolutions and temporal coverge. Many current studies have achieved 1km resolution products and the scale of daily even hourly.

**Response:** Thank you for your comment. We are unaware of any existing products of daily air temperature at the 1-km resolution with a global coverage. We believe our dataset is the first of this kind. We have clarified relevant description in the revised manuscript.

*"Our dataset aims to provide the first ever 1-km resolution daily maximum and minimum Ta dataset with a global coverage" (lines 79-80)*

**Comment #2-3**: line 103-104: As you trained models for each day of the period 2003-2020 as well as for each of the five regions, you would got the same number of results from 10-fold cross validation with trained models. Because different models will produce different validation results, how did you got only one result for each region, each landtype, each climate type, each year.... Possibly, you validate the products using all the records, thus, the validation is not independent. Or you caculated the average of RMSE of all models' 10-fold cross validation, such as in one region in one day, thus the RMSE through the paper is debatable.

**Response:** Thank you for your comment. Our cross-validation is a very rigorous and widely used method for evaluating the accuracy. The uncertainty estimates are based on independent data (not used in the model training). Moreover, our RMSE is a conservative estimate of the true uncertainties of our data because we use all available data, more than those (90%) in the cross-validation. In order to evaluate the model performance for estimating the gridded Ta, we used the 10-fold cross-validation for each day in each region. That is, we equally and randomly divided the data into 10 groups. Nine groups were used as training set and the rest one group was used as the testing set. This process was implemented for 10 times until all the groups were used as the testing set. Each test of the 10-fold cross-validation can generate a RMSE and the average RMSE of the 10 tests was used as the final RMSE for a specific day in a specific region. Therefore, the accuracy assessment was implemented based on independent validation data and can provide a reliable evaluation of the accuracy using the 10-fold cross-validation. We compared daily RMSEs based on 10-fold cross-validation and validation with 30% randomly selected testing data (Fig. R1). We found that the two validation methods show similar results in accuracy. The distributions of RMSEs using the 10-fold crossvalidation method is more concentrated with higher maximum densities than those of the 30%-random-validation-based method, especially in Africa and South America due to the low number of validation records, indicating the superiority of the 10-fold cross-validation method in stability.

For each station, we can calculate RMSE based on the time series of estimated and validation Ta in the 10-fold cross-validation. Accordingly, we can calculate mean RMSE and corresponding standard deviation in each land cover type, climate type, and elevation range. We have improved relevant descriptions in the revised manuscript.

*"The model performance for estimating gridded Ta was assessed based on root mean square error (RMSE) and mean square error (MAE) using the 10-fold cross-validation in these regions in each day. Taking the RMSE as an example, a RMSE was generated in each test of the 10-fold cross-validation and all RMSEs from the 10 tests were averaged as the final RMSE in a specific day in a specific region. This accuracy assessment using the 10-fold cross-validation was implemented based on independent validation data and can provide a reliable evaluation of the accuracy. For each station, we can also calculate RMSE based on the time series of estimated and validation Ta from the 10-fold cross-validation. Accordingly, we can calculate mean RMSE and corresponding standard deviation in each land cover type, climate type, and elevation range. Specifically, this accuracy assessment represents conservative estimates of the uncertainties of our data because when producing the final results, we used all the available data, more than those in the 10-fold cross-validation" (lines 134-143)*

[Figure]

*Figure R1: Density of daily RMSEs from 30%-random-validation (red) and mean RMSEs from 10-fold cross-validation (blue) in five regions in year 2010.*

**Comment #2-4**: Please describe how many records you filled because of missing values of weather station observations, and what percent of filled records to all records. Please note it's 'records', not number of sations with valid observations. Figure S1 and S3 would confuse readers because Y axis and title are inconsistent. Besides, how do you define valid station number? For example, Latin America in 2020 has no

more than 150 stations before filling missing data. After filling, it is over 150. If one station had only one valid record in 2020, it didn't make sense.

**Response:** Thank you for your suggestion. The number of filled records for Tmax and Tmin in Africa is 335,900 and 430,652, respectively, accounting for 17.7% and 21.4% of the total records. In Latin America, they are 469,637 and 348,554 for Tmax and Tmin, respectively, accounting for 29.4% and 19.4% of the total records. Based on the daily number of valid records, we drew a boxplot in each year. We have clarified it in the revised supplement.

*"Figure S1: Number of valid records (Y-axis) in five regions. Each box is based on the daily number of valid records in a specific year. Each point represents the number of valid records in a specific day."*

*"Figure S3: Number of valid records (Y-axis) after filling missing values in Africa and Latin America. Each box is based on the daily number of valid records in a specific year. Each point represents the number of valid records in a specific day. The number of filled records for Tmax and Tmin in Africa is 335,900 and 430,652, respectively, accounting for 17.7% and 21.4% of the total records. In Latin America, they are 469,637 and 348,554 for Tmax and Tmin, respectively, accounting for 29.4% and 19.4% of the total records."*

**Comment #2-5**: In equation (1), there is only one LST, however, as both Terra and Aqua observes in one daytime and nightime, there are two LST values. Did you use average value? or lower value? or higher value?

**Response:** Thank you for your questions. Observations from both Terra and Aqua satellites were used to build gap-filled LST by using a spatiotemporal gap-filling algorithm. More details can be found in Zhang et al. (2022a). We have clarified it in the revised manuscript.

Zhang, T., Zhou, Y., Zhu, Z., Li, X. and Asrar, G. R.: A global seamless 1 km resolution daily land surface temperature dataset (2003–2020), Earth Syst. Sci. Data, 14(2), 651–664, doi:https://doi.org/10.5194/essd-14-651-2022, 2022a.

*"The LST dataset is a global seamless 1-km resolution LST dataset at a daily (mid-daytime and mid-nighttime) frequency from 2003 to 2020, which was gap-filled from the MODIS LST products (Zhang et al., 2022a)." (lines 94-96)*

*re available over large areas." (lines 290-291)*

---

## Author Comment (AC3)

**Reviewer #3**

A global dataset of daily near-surface air temperature at 1-km resolution (2003-2020)

Review Comments

General comments:

Near-surface air temperature (Ta) has extensive applications in climate and environment studies. This study, based on a newly developed Spatially Varying Coefficient Models with Sign Preservation (SVCM-SP) algorithm, generated a global dataset of daily maximum and minimum Ta (Tmax and Tmin) at 1 km from 2003 to 2020 by integrating ground Ta observations from weather stations and gridded LST and DEM data. The assessment shows that the employed algorithm can effectively capture the negative relationships between Ta and elevation and the positive relationships between Ta and LST. The cross-validation indicates the estimated Ta show satisfactory accuracies, and the RMSEs of Ta estimates range from 1.20 to 2.44 ºC for Tmax and 1.69 to 2.39 ºC for Tmin.

The study designed a global maximum and minimum Ta estimation scheme and developed an applicable time-series (2003-2020) daily Ta dataset. I think this work is important because the generated datasets are of great demand and value in practical applications (e.g., urban climate research). However, some issues in the manuscript still need to be addressed before being ready for publication. The specific comments are given as follows.

**Response:** Thank you very much for your comments and suggestions. Below please find our responses to your comments in detail.

Specific comments:

**Comment #3-1**: Line 18, Please add the unit for '2.44'.

**Response:** Done.

**Comment #3-2**: Line 20, There is ambiguity in the expression. The positive and negative relationship is suggested to be expressed separately.

**Response:** Thank you for pointing it out. We have improved the description in the revised manuscript.

*"Our dataset correctly represents a negative relationship between Ta and elevation and a positive relationship between Ta and land surface temperature" (lines 20-21)*

**Comment #3-3**: Line 30, Why is the LST mentioned here?

**Response:** LST has been removed.

**Comment #3-4**: Line 70, What does 'these' refer to?

**Response:** Thank you for your question. We have clarified it in the revised manuscript.

*"To overcome such drawbacks, we recently proposed a class of Spatially Varying Coefficient Models with Sign Preservation (SVCM-SP) (Kim et al., 2021; Zhang et al., 2022b), which can capture and preserve relationships between Ta and explanatory variables." (lines 67-69)*

**Comment #3-5**: Line 88, It is recommended to add the region name represented by each color boundary, or to label the region name directly in Figure 1.

**Response:** Thank you for your suggestion. We have explained the name of regions in the caption of Figure 1 in the revised manuscript.

*"Specifically, polygons of red, purple, orange, blue, and black represent the boundaries of North America, Latin America, Africa, Australia, and Europe & Asia, respectively." (lines 90-91)*

**Comment #3-6**: Line 90, Do these ground measurements provide hourly Ta observations?

**Response:** Thank you for your question. No, the ground measurements we obtained only contain maximum and minimum Ta observations in each day. We clarified it in the revised manuscript.

*"Ground station-based Tmax and Tmin were compiled from a total of 103,156 weather stations from 2003 to 2020." (line 92)*

**Comment #3-7**: Line 96, What are the DEM data years used?

**Response:** Thank you for your question. We clarified it in the revised manuscript.

*"The DEM layer we used is the SRTM30_PLUS product at 1-km resolution (Becker et al., 2009), which has been generated from the combination of the Shuttle Radar Topography Mission (SRTM30) topography (collected in 2000) (Hennig et al., 2001; Rosen, 2000) within a latitude of ±55 degrees, ICESat derived topography (collected from February 1st, 2003 to June 30th, 2005) (Dimarzio et al., 2007) in Antarctica, and the GTOPO30 topography (completed in late 1996) (Danielson and Gesch, 2011) in the Arctic." (lines 96-100)*

**Comment #3-8**: Line 136-137, What does this sentence mean?

**Response:** Thank you for your question. This sentence means the interannual variations of the accuracy are smaller than spatial variations of the accuracy across regions. We have clarified it in the revised manuscript.

*"Meanwhile, the variation of accuracy across years in each region is smaller compared to the spatial variation of accuracy across regions (Tables S1-S2)." (lines 154-155)*

**Comment #3-9**: Line 216-227, There are too many introductions about previous studies, which are already discussed in the Introduction. It is suggested to simplify these contents.

**Response:** Thank you for your suggestion. We have simplified the description in the revised manuscript.

*"The gridded Ta data in this study have advantages regarding spatiotemporal resolutions (i.e., 1-km and daily maximum and minimum) and its global coverage (Table S3). The spatial resolution of existing global Ta datasets with daily frequencies and long-term coverage is generally low (e.g., 0.25°) (Hersbach et al.,*

*2018; Kalnay et al., 1996). Ta datasets with improved spatial resolutions (e.g., 1 km) are usually only available at the continental or national scales (Chen et al., 2021; Fang et al., 2021; MacDonald et al., 2020; Oyler et al., 2015; Thornton et al., 2021)." (lines 239-243)*

**Comment #3-10**: Line 227-239, Some contents (e.g., Line 228-232) that have been mentioned similarly in the Introduction are also suggested to be simplified.

**Response:** Thank you for your suggestion. We have simplified relevant descriptions in the revised manuscript.

*"The gridded Ta in this study can effectively capture the spatial variation of Ta by preserving physical relationships between Ta and response variables (Fig. S7). In other Ta datasets, such physical relationships (e.g., positive relationship between Ta and LST) cannot always be preserved in some situations because these datasets were created using methods without explicit constraints on the relationships between Ta and response variables. Efforts have been made to build vertical lapse models to estimate gridded Ta according to Adiabatic Lapse Rate (ALR) (Dodson and Marks, 1997; Rhee and Im, 2014; Thornton et al., 2021; Zhu et al., 2017), but the generalization of these models is limited because it is difficult to accurately capture ALR due to its spatial change." (lines 244-249)*

**Comment #3-11**: Line 228, What does the "which is not always true in other gridded Ta datasets" mean?

**Response:** Thank you for your question. It means that, in most cases, there is a negative relationship between elevation and Ta and a positive relationship between LST and Ta in other gridded Ta datasets. However, in some cases, there are opposite relationships. We have clarified it in the revised manuscript.

*"In other Ta datasets, such physical relationships (e.g., positive relationship between Ta and LST) cannot always be preserved in some situations because these datasets were created using methods without explicit constraints on the relationships between Ta and response variables." (lines 245-247)*

**Comment #3-12**: Line 244, Why is the UHI effect mentioned here? The previous description of the Ta dataset does not seem to refer to UHI.

**Response:** Thank you for your question. We mentioned the UHI effect for a weakness of Daymet in capturing the spatial variation of Ta in urban areas, although its overall accuracy is comparable to our dataset. We have clarified it in the revised manuscript.

*"Therefore, Daymet has difficulties in capturing the spatial variation of Ta in urban areas, although its accuracy is comparable to our dataset." (lines 256-257)*

**Comment #3-13**: Line 278-281, This is more appropriate for the discussion section than for the conclusion.

**Response:** Thank you for your suggestion. We have moved these descriptions to strengthen the discussion section, and simplified the description in the revised manuscript.

*"Second, we only used two covariates in the SVCM-SP algorithm although the potential of generalization of our framework is large. Additional covariates (e.g., other surface characters such as GLAS-derived*

*canopy height and vegetation parameters) can be explored in the SVCM-SP algorithm to further improve the model performance." (lines 268-270)*

*"Future work can focus on improving the accuracy of the gridded Ta dataset using the SVCM-SP algorithm by exploring more explanatory variables which are available over large areas." (lines 290-291)*

---

## Author Response (AR2)

**Reviewer #1**

I have checked the response submitted by the author, and most problems have been solved. But I still have some problems to solve before the manuscript is accepted and published (Minor revision)

**Comment #1-1**: The overlapping region was solved by the authors in the revised manuscript, but why the statistical accuracy in Table 2 did not change, especially in the South America region? According to the author's description, the model of each region is independent. Compared with the original paper, the number of stations in the South America region is significantly reduced, but the statistical results of Tmax and Tmin in South America in Table 1, Figure 4, and Figure 5 are not changed, which makes me very confused.

**Response:** Thank you very much for your comments. There are some changes related to South America. Accordingly, we have updated Tables 1, S1, S2, Figures 4, 5, S1, S3, and S5 at the regional level in the revised manuscript and supplement. However, the statistical accuracy in Tables 2 – 4 at the global level did not change, because there are only a few small changes in South America and the number of records for validation in South America are very small compared to the global total.

**Comment #1-2**: I don't agree with the response to my comment 9. I don't think the 10-fold cross validation method in stability has superiority over independent validation. Although the training and test sets of 10-fold cross validation are independent, the correlation between neighboring stations is often ignored for regions with a high density of ground-based stations, so the difference between the results of 10-fold cross validation and random sampling validation is small in regions with high station density (e.g., North America). At the same time, there are some differences in the validation results in South America and Australia, where the station density is relatively small. Given the spatial correlation between stations, the independent validation methods for different years of data as training set and validation respectively (such as the data from 2003-2018 as the training set, and the data in 2020 as the validation set) can more reasonably portray the retrieval accuracy of Tmax and Tmin in the pixel region of the missing ground-based stations.

**Response:** Thank you very much for the suggestion. We used the 10-fold cross-validation in this study for two main reasons. First, 10-fold cross-validation is more reliable compared to the random sampling validation (Figure R1), especially for regions with a low station density. Both methods were implemented based on independent random sampling and do not have large differences except for the number of evaluations. In this experiment (Figure R1), the result shows that the accuracies vary largely across five evaluations using the random sampling method. The differences in accuracy between the two validation methods and also across the evaluations using the random sampling method were mainly caused by their sensitivities to station densities. The accuracy assessment results are more stable in regions with a higher station density (e.g., North America) because more stations were used in both the 10-fold cross-validation and the suggested random sampling method. Compared to the random sampling method, our 10-fold cross-validation is more reliable, especially in regions with a low station density.

Second, the suggested validation method using data from a different year is not applicable to our method in this study because spatial correlations between Ta and explanatory variables were not assumed constant across days and years. We agree that some models (e.g., random forest and deep learning) in literature have implemented temporal evaluations of model performance due to their underlying assumptions on fixed spatiotemporal correlations. However, in this study, our model is similar as a spatial interpolation technique.

It is not applicable to assume the same spatial correlations between Ta and explanatory variables across time. For example, there is no theoretical basis to find two corresponding days from different years as training and testing days. Therefore, we fitted the SVCM-SP model and estimated the gridded Ta using data in the same day.

[Figure]

*Figure R1: Comparison of RMSEs (top: Tmax; bottom:Tmin) in Africa in 2010 between the five evaluation using the random sampling method (red) and 10-fold cross-validation (blue).*

**Comment #1-3**: The regression line in Figure 3 is incorrect, for example, the regression equation between the estimated and measured values of Tmax over the African region is Y=1X, which means that the estimated Tmax and measured Tmax are exactly the same, which is obviously inconsistent with the value of R (not equal to 1) and the scatter plot, so the form of the regression equation should be modified to "Y=aX+b"

**Response:** Thank you for your suggestion. As suggested, we have revised Figure 3 below.

[Figure]

*Figure 3: Scatter plots between estimated and observed Ta in five regions in year 2010. Each point represents the estimated and observed Ta (Tmax or Tmin) in a specific day in a weather station. The color of points represents the density, in which red and blue points represent the high and low densities, respectively. The red line is the regression line and the black line is the 1:1 line.*